# Assessment of the Genetic Characteristics of a Generation Born during a Long-Term Socioeconomic Crisis

**DOI:** 10.3390/genes14112064

**Published:** 2023-11-11

**Authors:** Svetlana V. Mikhailova, Dinara E. Ivanoshchuk, Pavel S. Orlov, Ahmad Bairqdar, Maksim S. Anisimenko, Diana V. Denisova

**Affiliations:** 1Institute of Cytology and Genetics, Siberian Branch of Russian Academy of Sciences (ICG SB RAS), 10 Prospekt Ak. Lavrentyeva, 630090 Novosibirsk, Russia; dinara@bionet.nsc.ru (D.E.I.); orlovpavel86@gmail.com (P.S.O.); bairqdar@bionet.nsc.ru (A.B.); m.anisimenko@yahoo.com (M.S.A.); 2Institute of Internal and Preventive Medicine—Branch of ICG SB RAS, 175/1 Borisa Bogatkova Str., 630089 Novosibirsk, Russia

**Keywords:** stress, genetic predisposition, polymorphism, *SLC6A3*, *MAOA*, *CRHR1*, *FKBP5*, *OXTR*, *CNR1*, *ACE*, *CHRNA4*, *ANKK1*, *APOE*

## Abstract

Background: A socioeconomic crisis in Russia lasted from 1991 to 1998 and was accompanied by a sharp drop in the birth rate. The main factor that influenced the refusal to have children during this period is thought to be prolonged social stress. Methods: comparing frequencies of common gene variants associated with stress-induced diseases among generations born before, after, and during this crisis may show which genes may be preferred under the pressure of natural selection during periods of increased social stress in urban populations. Results: In the “crisis” group, a statistically significant difference from the other two groups was found in rs6557168 frequency (*p* = 0.001); rs4522666 was not in the Hardy–Weinberg equilibrium in this group, although its frequency did not show a significant difference from the other groups (*p* = 0.118). Frequencies of VNTRs in *SLC6A3* and *MAOA* as well as common variants rs17689918 in *CRHR1*, rs1360780 in *FKBP5*, rs53576 in *OXTR*, rs12720071 and rs806377 in *CNR1*, rs4311 in *ACE*, rs1800497 in *ANKK1*, and rs7412 and rs429358 in *APOE* did not differ among the groups. Conclusions: a generation born during a period of prolonged destructive events may differ from the rest of the gene pool of the population in some variants associated with personality traits or stress-related disorders.

## 1. Introduction

The main approach to studying natural selection in modern human populations that have undergone a demographic transition (a decrease in infant mortality followed by a decrease in fertility) is assessing the association between the fertility rate and polymorphic variants in the genome. Standard methods of medical genetics (analysis of genotype–phenotype correlations) cannot be used to analyze the direction of selection because most diseases—especially in the postreproductive period—are not directly related to the number of children an individual has. Large genome-wide association studies (GWASs) of fertility (the number of children ever born) allow the identification of groups of genes subject to natural selection at present and in recent history. It has been shown that these are mainly genes determining the hypothalamic–pituitary–gonadal axis, body measurements, behavioral traits, and social cognition [1,2,3]. Another approach is to compare allele frequencies among several successive generations of modern humans, for example, the Framingham Study [4] or ancient humans. There are only a few studies about the influence of prolonged destructive events on the genetics of generations born at those times. In a recent GWAS on infant mortality [5], signs of directional selection were found in the generation born during the period of maximum German bombing (World War II) in England and Wales. Among other sites are at the following loci: rs1446585 of the *R3HDM1* gene near the *LCT* gene (lactose tolerance), rs5743618 of the *TLR1* gene (innate immunity), and rs2852853 in an intron of the *DHCR7* gene (vitamin D metabolism). At loci *LCT* and *TLR1/6/10*, directional natural selection of various intensity levels has been identified over several millennia among Europeans [6]. This apparently intensified during the war period. Those authors attributed this finding to poor living conditions.

Several studies have been carried out using DNA isolated from human remains from burial sites. Comparing allele frequencies between burials of different ages and modern populations makes it possible to identify genes that have been under selection pressure in the past. In time series data from Poland (1st–18th centuries AD), polymorphic sites in the six innate-immunity genes that are associated with predisposition to mycobacterial infections have been analyzed [7]. Signs of selection were found there for rs17235409 of the *SLC11A1* gene and for rs1800896 of the *IL10* gene; this phenomenon may be due to the spread of tuberculosis in Europe. Three SNVs in noncoding regions (rs11571319, rs17473484, and rs1052025) and one intronic rs2549794 show a significant difference in frequency when DNA is compared between burial sites set up before, during, and after the bubonic plague epidemic in Britain and Denmark [8].

Previously, in a pilot study, we compared frequencies of variants associated with stress resilience and stress-induced diseases among groups of adolescents born before, after, and during the prolonged socioeconomic crisis in Russia in the 1990s [9]. According to statistics, during the 1990–2000 period, the birth rate decreased from 13.4 to 8.7 per 1000 people overall in Russia, and from 13.2 to 8.5 in Novosibirsk Oblast [10]. During this period, along with an almost twofold drop in industrial and agricultural production, the population was affected by increased psychosocial pressure, weakening social support, and sharp changes in the scale of values, and these factors caused a decrease in social well-being and an increase in social stress [11]. Social stress is known to affect fertility rates in humans [12,13], and predisposition to stress-induced disorders is inherited [14]. It can be theorized that under stress conditions, the degree of the decline in fertility differed among people depending on their genotypes. In this context, the generation born during a crisis may differ in the frequency of some genetic variants from the generation that was at a reproductive age during the crisis in the same population, as well as from the generations born immediately before and after the crisis. We hypothesized here that such differences in allele frequencies can be found in genes predisposing individuals to stress-induced disorders or stress resilience. Previously, a statistically significant difference among the “crisis” and control groups has been found in the frequencies of the long (9R + 10R) allele of the VNTR in exon 3 of *DRD4* and rs4680 G of *COMT* [9]. Both genes belong to the dopaminergic regulatory system. In the current study, we compared among the same groups of adolescents frequencies of variants of several other genes that are known to be associated with stress-induced disorders. The prevalence of single-nucleotide variants (SNVs) and VNTRs of the genes of the dopaminergic system, hypothalamic–pituitary–adrenal axis (HPA), neurotransmission, hormone receptors, and metabolic genes was estimated.

*DRD2*, *DAT1*, *CHRNA4*, and *MAO* are involved in dopaminergic pathways directly or indirectly. The *MAOA* (monoamine oxidase A) gene encodes the mitochondrial enzyme for the deamination of dopamine, norepinephrine, and serotonin. The gene is located on the X chromosome, and therefore males (hemizygotes) are studied more often. The VNTR in this gene’s promotor (1.2 kbp upstream of exon 1) consists of a 30-bp sequence (R) repeated 2, 3, 3.5, 4, 5, or 6 times. The number of the repeats affects the *MAOA* transcriptional activity [15]. *MAOA* alleles have accordingly been classified as MAOA-H (high: 3.5R, 4R, and 5R) or MAOA-L (low): 2R and 3R [16]. Data on the association of the *MAOA* VNTR with stress-induced diseases and personality traits are contradictory, and depend on sex and on gene–environment (G × E) interaction. It has been suggested that individuals with MAOA-L may be at increased risk of depletion of an HPA response under stress [17]. Male carriers of MAOA-H have a larger dopamine release and elevated aggression after a violent movie [18], whereas female carriers have higher scores on both burnout and depression after stressful life events [19]. In female 3R carriers, there is higher emotional stability, which negatively correlates with depression and anxiety [20]. Among Asians, higher frequencies of 4R/4R (females) and 4R (males) genotypes has been observed in a group affected by major depressive disorder (MDD) as compared to healthy controls [21].

The *SLC6A3 (DAT1*) gene encodes the dopamine transporter, which controls concentrations of dopamine by reuptaking dopamine from the synapse. The 3′ untranslated region (3′-UTR) VNTR of a 40-bp element has been identified to contain more common versions 9R and 10R and rarer versions 3R, 5R, 7R, 8R, and 11R [22]. Data on the effect of carriage of different alleles on the transcriptional activity of this gene are contradictory, but according meta-analyses, the 9R allele is associated with a higher gene activity [23,24]. There is also contradictory evidence for the association of this VNTR with stress-induced disorders and temperament traits [25,26,27]. It is believed that the age–sex–G–E interaction and the genomic environment both within and between genes, including the influence of race, play an important role in the formation of the phenotype [28].

Rs1800497 C>T (originally polymorphism TaqIA A1/A2) is a missense variant in exon 8 of the *ANKK1* gene (ankyrin repeat and kinase domain-containing 1: a Ser/Thr protein kinase involved in signal transduction pathways). This SNV is often assigned to dopamine receptor gene *DRD2*, the 3′ end of which is located 9.5 kbp away from this site [29]. It has been demonstrated that rs1800497 influences striatal D2 receptor availability [30] and *DRD2* expression [31], but *ANKK1* itself may function in dopaminergic pathways [32]. Rs1800497 T is reported to be a risk allele for mood disorders, addictions, and post-traumatic stress disorder (PTSD) [33,34].

*CHRNA4* (cholinergic receptor nicotinic α 4 subunit) encodes α4 subunit of nicotinic acetylcholine receptors. CHRNA4 is a ligand-gated ion channel situated in the postsynaptic membrane. It modulates the release of dopamine, serotonin, γ-amino butyric acid, and glutamate [35]. Among 84 polymorphic sites in genes that influence catecholamine function or are associated with risk taking, it is rs4522666 (in the 3′ region of *CHRNA4*) that is associated with higher harm avoidance (HA) scores on the Temperament and Character Inventory [36]. HA is a personality trait characterizing the intensity of a subject’s reaction to aversive stimuli, and correlates with depression and anxiety [37].

*FKBP5* (FK506-binding protein 51) encodes an inhibitor of glucocorticoid receptor (GR) transcriptional activity and is involved in the regulation of the HPA. FKBP5 and GR participate in a negative feedback loop within HPA, wherein FKBP5 strongly inhibits GR activity by lowering its nuclear translocation, whereas GR controls *FKBP5* transcription [38]. Intronic variant rs1360780 is located close to a functional glucocorticoid response element [39]; carriage of the T allele causes *FKBP5* overexpression via stronger *FKBP5* mRNA induction after GR activation [40]. It is believed that carriage of this allele during chronic psychosocial stress may disturb HPA regulation [41]. The rs1360780 T allele correlates with MDD and higher anxiety scores [42,43], and its C allele with PTSD [44].

*CRHR1* (corticotropin-releasing hormone receptor 1) codes for a major HPA regulator. The binding of corticotropin-releasing hormone (CRH) to CRHR1 triggers the release of cortisol, the main stress response hormone. The CRH–CRHR system is involved in mood and anxiety disorders [45]. In the cerebellum, the minor A allele of rs17689918 results in significantly reduced *CRHR1* expression [46]. Carriage of the A allele raises the risk of panic disorder in women, and under-expression is thought to cause a phenotype characterized by persistent fear [47]. The association of rs17689918 genotypes with symptoms of mood and anxiety disorders depends on sex and the number of stressful events experienced in a lifetime [48].

The *OXTR* (oxytocin receptor) gene encodes a G protein–coupled receptor for oxytocin, which is a posterior pituitary hormone. The oxytocin signaling pathway takes part in the modulation of behaviors, including stress-induced activities, and is associated with psychiatric diseases such as mood disorders [49]. Data on the association of the intronic rs53576 polymorphism with stress-related disorders are contradictory [49]. In nonclinical populations, the effects of the carriage of different genotypes of rs53576 are race-specific [50,51].

The *ESR1* (estrogen receptor 1) gene encodes a nuclear receptor regulating the transcription of estrogen-inducible genes that play a part in metabolism and reproductive functions and regulate stress response [52]. It has been shown that some *ESR1* genotypes and haplotypes predispose individuals to mood disorders and anxiety [53]. *ESR1* is among the leading causal genes mostly underlying MDD and PTSD phenotypes [54]. In a GWAS on anxiety, intronic rs6557168 C>T proved to be the second strongest genome-wide significant independent signal among US military veterans [55]. Risk allele C had the same direction of effect in another GWAS, performed on a Danish population [56].

*CNR1* (cannabinoid receptor types 1) codes for G protein–coupled receptor CB1R. It is ubiquitously expressed in the central nervous system and in peripheral neurons. Endocannabinoid signaling participates in neural plasticity, learning, and regulation of HPA response to acute or repeated stress. It is believed that the mechanism of development of stressor tolerance is based on the initiation of CB1R signaling caused by repeated stress, thus allowing the risk of negative consequences to be reduced [57,58]. 3′-UTR SNV rs12720071 and intronic SNV rs806377 are located in different haplotype blocks within *CNR1* [59]. Variant rs12720071 G in *CNR1* confers sex-specific susceptibility to panic disorder [60]. Carriage of rs806377 C presumably correlates with higher social reward responsivity [61].

*APOE* (apolipoprotein, apoE) encodes a major apoprotein of the chylomicron family. It has been suggested that *APOE*, along with *FKBP5*, *OXTR*, and *CRHR1*, is among the main genes associated with resilience across divergent resilience definitions [62]. Stress induces apoE synthesis as part of repair processes in the central nervous system [63]. The structure of human apoE is determined by two common exonic variants (rs7412 and rs429358) resulting in Arg > Cys substitutions. Originally, three discovered alleles had been designated ε2, ε3, and ε4. It is thought that apoE ε4 is susceptible to proteolysis, giving rise to neurotoxic fragments in neurons [63]. Several articles have confirmed reduced neuronal resilience in carriers of the ε4 allele and higher resilience in carriers of the ε2 allele during stress response [64,65]. Moreover, resilience is more related to a patient’s *APOE* genotype than to the etiology of stress. Among military veterans, ε4 allele carriage correlates with PTSD symptom severity [66,67]. Opposite results were obtained in an assessment of subclinical symptoms of anxiety during a 19-day military training; *APOE* ε4 noncarriers showed significant changes in anxiety and depressive symptoms and in their sense of coherence, unlike ε4 carriers [68]. It is possible that an *APOE* genotype interacts differently with acute and chronic stress.

*ACE* (angiotensin I-converting enzyme) encodes an enzyme located at the cell surface and hydrolyzing circulating peptides. It takes part in fluid and electrolyte balancing, blood pressure regulation, and vascular remodeling [69], and mediates stress and fear responses by influencing HPA activity [70,71]. The T allele of rs4311 is reported to be associated with PTSD [72] and syndromal panic attacks [73].

In this paper, we compare the frequencies of the aforementioned SNVs and VNTRs among group of adolescents born during a long-term socioeconomic crisis in Russia (in the 1990s) and two control groups: those born before and after this crisis.

## 2. Materials and Methods

The study protocol was approved by the ethical committee of the Institute of Internal and Preventive Medicine, a branch of the Institute of Cytology and Genetics of the Siberian Branch of the Russian Academy of Sciences, Novosibirsk, Russia (protocol No. 7, approved on 22 June 2008). Blood samples were obtained during a standardized medical cross-sectional examination, which has been carried out since 1989 every 5 years by the Institute of Internal and Preventive Medicine (Novosibirsk, Russia). These adolescent groups have been described previously [74]. Random representative groups of unrelated schoolchildren aged 14–17 years at a male–female ratio of 42:58 were formed in randomly selected grades of 10 secondary schools in the Oktyabrsky district of Novosibirsk city, which is a typical area in the industrial center of Western Siberia. A complete survey of these schoolchildren was conducted, and the response rate was 95%. During each examination, blood samples were collected from approximately 10% of adolescents living in the study area, and this approach ensured the representativeness of the population sample.

We analyzed adolescent subjects’ blood samples collected in the years 1999, 2009, and 2019. An additional group of adolescents’ blood samples from 2014 was used for rs4522666 genotyping. Accordingly, 4 independent groups of adolescents were compiled: those born in 1982–1985 (group 1); those born in 1992–1995 (the period of the socioeconomic crisis of the 1990s in Russia: group 2); subjects born in 2002–2005 (group 3); and adolescents born in 1996–2000 (the period of end of the crisis: group 4). Each group consisted of at least 95% Caucasian people; the ethnicity of the individuals was determined using a questionnaire and an additional cross-sectional survey to identify the nationality of the ancestors.

Blood samples were stored at −20 °C until genomic DNA was isolated from blood leukocytes via standard phenol–chloroform extraction [75]. For PCR analysis of DNA fragments of genes *MAOA* and *SLC6A3*, master mix BioMaster HS-TaqPCR (2×) (Biolabmix, Novosibirsk, Russia) and direct and reverse primers (Biosset, Novosibirsk, Russia) were employed, as shown in Appendix A.

Genotyping of rs429358 and rs7412 was performed according to [76]. The sizes of the PCR products and of restriction fragments were estimated via electrophoresis in a 5% polyacrylamide gel.

Genotyping of rs17689918, rs1360780, rs53576, rs12720071, rs806377, rs6557168, rs4311, rs4522666, and rs1800497 was conducted using real-time PCR on a LightCycler 96 instrument (Roche, Basel, Switzerland). Flanking oligonucleotides were selected in Primer-BLAST. TaqMan probes were designed based on a gene sequence retrieved from the NCBI in Vector NTI Advance 11.0 software (Thermo Fisher Scientific, Waltham, MA, USA). The flanking oligonucleotides and complementary probes labeled with dyes FAM and HEX and with the BHQ1 fluorescence quencher are listed in Appendix A.

The RT-PCR was carried out in a 25 μL reaction mixture consisting of 15 ng of DNA, 100 nM forward and reverse flanking primers (10 μM), 200 nM each of the 2 probes, and master mix BioMaster HS-qPCR (2×) (Biolabmix, Novosibirsk, Russia) according to manufacturer’s protocol. The amplification program began with denaturation at 95 °C for 300 s, followed by 45 cycles of 95 °C for 10–15 s and T_m_ for 30 s. Next, in the 2 multi-well plates analyzed first for each polymorphic site, 10% of the DNA samples were verified via Sanger sequencing. The obtained data were processed with LightCycler 96 software version 1.1.0.1320.

Statistical analysis consisted of an intergroup comparison of allele frequencies for each of the studied polymorphisms by Fisher’s exact two-tailed test using SPSS 11.0 software (IBM Corp., Armonk, NY, USA). In the case of multiple alleles, the following comparisons were made: for 3′-UTR VNTR *SLC6A3* (9R vs. all others), for VNTRs in the promotor region of *MAOA* (2R + 3R vs. all others), and for *APOE* (ɛ2 vs. all others and ɛ4 vs. all others).

## 3. Results

The results of genotyping for rs17689918, rs1360780, rs53576, rs12720071, rs806377, rs6557168, rs4311, rs4522666, rs1800497, rs7412, and rs429358 are presented in Table 1.

The Hardy–Weinberg equilibrium (HWE) was noted in all groups except the first control group (group 1) for rs17689918 of *CRHR1* (χ^2^ = 9.4617) and the “crisis” group (group 2) for the rs4522666 of *CHRNA4* (χ^2^ = 10.0955). In the combined control (groups 1 + 3), the HWE was not disturbed for rs17689918 (χ^2^ = 2.7559). The additional group (group 4; born in 1996–2000, i.e., at the end of the crisis)—compiled in 2014 within the same medical cross-sectional examination—was analyzed for rs4522666. It was in HWE (χ^2^ = 0.6675), but the summation of groups 2 and 4 increased the deviation from the HWE (χ^2^ = 10.1978) despite the same rs4522666 allele frequency among these groups (*p* = 0.766, χ^2^ = 0.092). Data on *APOE* alleles (ɛ2, ɛ3, and ɛ4) were combined; the results are shown in Table 2.

The results of genotyping for VNTRs in the promotor region of *MAOA* and in the 3′-UTR of *SLC6A3* are in Table 3 and Table 4. *MAOA* genotypes were estimated for males and females separately. No other variants described in the literature besides 8R, 9R, 10R, and 11R were found in *SLC6A3*.

An assessment of the statistical significance of the differences between the tested groups is given in Table 5.

There were no statistically significant differences in the frequencies of the tested VNTR variants of *SLC6A3* and *MAOA*, or in the frequencies of rs1800497, rs53576, rs12720071, rs806377, rs1360780, rs17689918, rs4311, and *APOE* ɛ2/3/4 among the adolescent groups.

In a combined control group of Russian adolescents, the frequency of the rs17689918 A allele (*CRHR1*) was significantly lower as compared to gnomAD data for non-Finnish Europeans (*p* < 0.001) and similar to the frequency described for Germans [47]. Apparently, the prevalence of this variant varies among European populations. Frequency of the rs6557168 C allele (*ESR1*) was significantly higher in the crisis group (*p* = 0.001) than in the controls.

## 4. Discussion

This study revealed a difference between the crisis group and control groups in 2 of 13 polymorphic sites in genes for which an association with stress-induced disorders or temperament traits has previously been described. The difference in the frequency of rs4522666 allele G (*CHRNA4*) between groups did not reach statistical significance (*p* = 0.176 for group 2 vs. groups 1 + 3 and *p* = 0.117 for groups 2 + 4 vs. groups 1 + 3), while the observed deviation from the HWE possibly indicates the “assortative” birth of children among the parents of the crisis group. For example, this might be possible if the birth rate decreased to a lesser extent among couples in which both parents had the rs4522666 GG genotype. The function of this SNV is unknown. Our results require verification through an analysis of large cohorts. In addition, it may be of interest to evaluate a possible association between the birth rate and HA scores in populations depending on social stress pressure, cultural values, and a novelty-seeking score, because the latter can modulate the effect of the HA score on reproductive behavior and social anxiety symptoms [79].

Notably, rs6557168 allele C (*ESR1*), whose frequency turned out to be elevated in the crisis group, has been shown to correlate with heightened anxiety among military veterans and in the Danish population [55,56]. Additionally, rs6557168 is associated with suicidal thoughts and behaviors among US military veterans [80]. The results obtained in this paper were unexpected, with both studies showing a correlation of rs6557168 with anxiety and also indicating a positive correlation of anxiety with fertility [55,56]. Six intronic SNVs of *ESR1* are reported to be predictors of attention deficit hyperactivity disorder (ADHD) severity [81]. Previously, using the same samples of adolescents, we have identified differences in the frequency of polymorphic variants in genes *DRD4* and *COMT*, which encode modulators of novelty processing and are associated with ADHD [9]. Higher polygenic scores of ADHD risk predict a higher fertility rate [82,83]. It is likely that some of the genetic variants that increase susceptibility to anxiety and ADHD are adaptive in the case of prolonged stress in a population. There is a substantial genetic overlap between well-being and depressive symptoms. In a recent article on UK Biobank data, subtracting the “depressive symptoms” GWAS from the “happiness and meaning of life” GWASs revealed that ADHD and risk taking have a direct genetic correlation with well-being [84]. Perhaps such a subtractive approach in the future will enable researchers to separate the adaptive and maladaptive genetic components of increased anxiety. There is currently no information about functional significance of the rs6557168 intronic locus. Signals detected in GWASs are not always related to the nearest locus or even gene, and replication of the results in other cohorts as well as linkage and functional analyses are necessary to identify the causative variant.

## 5. Conclusions

Thus, as a result of comparing groups of adolescents born in different periods relative to the socioeconomic crisis, it is demonstrated here that a generation born during a period of elevated social stress may differ from control groups (born before and after this period) in the frequency of alleles and genotypes of genes associated with stress-induced disorders and temperament dimensions. It is not known whether the changes observed in the crisis group reflect general trends of human evolution owing to increasing social stress caused by population density growth and urbanization, or whether such crisis events temporarily change the direction of adaptation in the population.

## 6. Limitations

The format of our study does not allow us to establish which of the variants came to each subject from the father and which from the mother. Some of the genes studied have sex-specific associations with personality traits and stress-induced diseases, so we lost some information. In addition, we cannot take into account all environmental factors that influenced the adolescents’ parents. SNVs rs4522666 and rs6557168 are located in no-coding regions, so it is difficult to predict their functional significance. Larger sample sizes are needed, especially for *MAOA* VNTR (due to splitting the cohorts by sex), rs17689918 (due to the Hardy–Weinberg imbalance in one of the control groups), and rs4522666 (to achieve statistical significance). Replication of these results in independent cohorts is necessary.

## Figures and Tables

**Table 1 genes-14-02064-t001:** Number of genotype carriers in groups of adolescents born in 1982–1985 (group 1), in 1992–1995 (group 2), and in 2002–2005 (group 3); an additional group born in 1996–2000 (group 4) was genotyped only for rs4522666.

**Group**	**rs17689918 *CRHR1***				**rs4311 *ACE***			
	GG	AG	AA	n	TT	TC	CC	n
1	432	68	9	509 ^§^	45	123	88	256
2	380	94	2	476	67	148	94	309
3	221	59	2	282	55	141	85	281
	**rs1360780 *FKBP5***				**rs1800497 *ANKK1-DRD2***			
	AA	AG	GG	n	GG	GA	AA	n
1	13	113	131	257	234	112	18	364
2	22	129	160	311	419	211	28	658
3	17	109	156	282	127	50	10	187
	**rs6557168 *ESR1***				**rs53576 *OXTR***			
	TT	TC	CC	n	GG	AG	AA	n
1	166	95	21	282	120	125	36	281
2	174	150	48	372	114	133	36	283
3	152	99	27	278	121	124	39	284
	**rs12720071 *CNR1***				**rs806377 *CNR1***			
	AA	AG	GG	n	AA	AG	GG	n
1	280	66	2	348	75	127	79	281
2	416	71	5	492	85	126	69	280
3	238	42	1	281	78	144	59	281
	**rs7412 *APOE***				**rs429358 *APOE***			
	CC	CT	TT	n	TT	TC	CC	n
1	313	50	3	366	274	93	3	370
2	474	83	4	561	398	129	10	537
3	315	58	5	378	368	88	6	462
	**rs4522666 *CHRNA4***							
	AA	AG	GG	n				
1	128	174	44	346				
2	258	288	133	679 ^§^				
3	136	190	50	376				
4	66	85	35	186				

^§^ groups with Hardy–Weinberg imbalance, n: an analyzed group’s size; allele definitions are according to the dbSNP database [77].

**Table 2 genes-14-02064-t002:** Numbers of *APOE* genotype carriers and allele frequencies in the groups of adolescents: born in 1982–1985 (group 1), born in 1992–1995 (group 2), and born in 2002–2005 (group 3); n: an analyzed group’s size.

*APOE*	Group 1, n = 362	Group 2, n = 533	Group 3, n = 360
**Number of genotype carriers**			
ɛ2/ɛ2	3	4	5
ɛ2/ɛ3	43	70	38
ɛ2/ɛ4	6	11	16
ɛ3/ɛ3	224	320	243
ɛ3/ɛ4	82	118	54
ɛ4/ɛ4	4	10	4
**Allele frequency**			
ɛ2	0.076	0.083	0.089
ɛ3	0.791	0.777	0.803
ɛ4	0.133	0.140	0.108

**Table 3 genes-14-02064-t003:** Numbers of genotype carriers and allele frequencies for VNTR in the promoter of *MAOA* in three groups of adolescents (1: born in 1982–1985, 2: born in 1992–1995, and 3: born in 2002–2005), R: the number of repeats in the VNTR, n: sample size (male/female).

*MAOA*	Group 1, n = 519 (227/292)	Group 2, n = 662 (280/382)	Group 3, n = 437 (168/269)
**Number of genotype carriers, males**			
2R	1	0	0
3R	93	101	64
3.5R	1	3	1
4R	129	175	101
5R	3	1	2
**Number of genotype carriers, females**			
2R/3R	0	1	1
2R/4R	3	0	0
3R/3R	35	66	43
3R/3.5R	2	1	1
3R/4R	133	151	102
3R/5R	3	3	0
3.5R/4R	6	3	1
4R/4R	103	150	118
4R/5R	7	6	3
5R/5R	0	1	0
**Allele frequency, males**			
2R	0.004	0	0
3R	0.410	0.361	0.381
3.5R	0.004	0.011	0.006
4R	0.568	0.625	0.601
5R	0.013	0.004	0.012
**Allele frequency, females**			
2R	0.005	0.001	0.002
3R	0.356	0.377	0.353
3.5R	0.014	0.005	0.004
4R	0.608	0.602	0.636
5R	0	0.014	0.006

**Table 4 genes-14-02064-t004:** Numbers of genotype carriers and allele frequencies for the VNTR in the 3′-UTR of the *SLC6A3* gene in three groups of adolescents (1: born in 1982–1985, 2: born in 1992–1995, and 3: born in 2002–2005). R: the number of repeats in the VNTR, n: sample size.

*SLC6A3*	Group 1, n = 328	Group 2, n = 660	Group 3, n = 373
**Number of genotype carriers**			
8R/10R	5	3	0
9R/9R	19	28	24
9R/10R	104	250	139
9R/11R	1	0	0
10R/10R	199	369	203
10R/11R	0	10	7
**Allele frequency**			
8R	0.007	0.002	0
9R	0.218	0.231	0.250
10R	0.772	0.758	0.740
11R	0.002	0.008	0.009

**Table 5 genes-14-02064-t005:** Frequencies of the studied alleles according to the Genome Aggregation Database (gnomAD [78]) for non-Finnish Europeans and the groups of adolescents (1: born in 1982–1985, 2: born in 1992–1995, and 3: born in 2002–2005), with a *p* value for an intergroup comparison of allele frequencies.

Polymorphic Site, Gene		Allele	Frequency				*p* Value			
			GnomAD	1	2	3	1↔2	2↔3	1↔3	2↔1+3
3′-UTR VNTR *SLC6A3*		9R	-	0.218	0.231	0.250	0.531	0.334	0.165	0.856
VNTR in promoter *MAOA*	male	2R + 3R	-	0.414	0.361	0.381	0.233	0.687	0.534	0.335
	female			0.361	0.378	0.355	0.532	0.414	0.852	0.381
rs1800497 *ANKK1-DRD2*		A	0.193	0.203	0.203	0.187	1.000	0.557	0.576	0.760
rs4311 *ACE*		T	0.469	0.416	0.456	0.447	0.185	0.770	0.324	0.335
rs6557168 *ESR1*		C	0.342	0.242	0.331	0.257	0.001	0.033	0.220	0.001 ***
rs4522666 *CHRNA4*		G	0.360	0.379	0.408 ^§^ (0.410 ^§^)	0.386	0.215 (0.169)	0.330 (0.266)	0.861	0.176 (0.117)
rs53576 *OXTR*		A	0.333	0.351	0.362	0.356	0.381	0.853	0.901	0.747
rs12720071 *CNR1*		G	0.090	0.101	0.082	0.078	0.225	0.846	0.199	0.498
rs806377 *CNR1*		A	0.489	0.507	0.471	0.466	0.233	0.905	0.189	0.569
rs1360780 *FKBP5*		A	0.288	0.270	0.278	0.254	0.790	0.357	0.533	0.460
rs17689918 *CRHR1*		A	0.196	0.084 ^§^	0.103	0.111	0.164	0.605	0.087	0.489
*APOE* ɛ2/3/4		ɛ2	0.077	0.076	0.083	0.089	0.596	0.730	0.390	0.942
		ɛ4	0.149	0.133	0.140	0.108	0.675	0.051	0.170	0.165

In parentheses: combined group 2 + 4 data; R: the number of repeats in a VNTR; ^§^ groups with a Hardy–Weinberg imbalance; *** the result is statistically significant.

## Data Availability

The data presented in this study are available upon request from the corresponding author.

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
