# Peer review of "Assessment of the Genetic Characteristics of a Generation Born during a Long-Term Socioeconomic Crisis"

_genes, 2023, doi:10.3390/genes14112064_

Round 1
Reviewer 1 Report
Comments and Suggestions for Authors
The authors have selected some interesting and obvious candidate genetic markers to compare their frequencies in adolescents born during long-term socio-economic crisis in Russia (1990s) vs. control groups those born before and after this crisis. I consider the cohort n# suitable and admire the enormous effort to maintain the relationship with this cohort.
I would have liked to see a control group, one outside of Russia as a comparison. The authors do use GnomAD as a comparison for allele frequencies (which are high).
I would recommend adding some references to define the “long-term socio-economic crisis in Russia” (Line 51).
I find the approach of selecting the candidate genes a bit bias and understand that there is probably lack of funding to support processing their cohort using genome sequencing or a very small pilot GWAS. Most of the genes that they are targeting are involved in many different genetic disorders.
The methdology used was well described and repeatable.
Medical evaluation of the human subjects was conducted every 5 years that included monitoring of mental, physical, and emotional state. There was more information regarding the ascertainment in their previous publication (Reference 7. Mikhailova et al., 2022). But , how does one collect honest emotional data over 5 year period, is there a regular questionnaire? Data from this is not shown in the paper regarding the mental, physical and emotional state. The phenotypic data needs to be declared; the range, average to show if the three groups differ in any way.
The authors have declared the issues with the paper and I am in agreement with that. However, what about the status of the genetic markers in animal models for anxiety and fertility?
Without any phenotypic data, I”m not convinced that this paper should be published in Genes. Based on the data shown, it is very subjective and open to bias.
Author Response
Dear Reviewer! Thank you very much for your attention to our article. We hope that we were able to improve it according to your comments
I would have liked to see a control group, one outside of Russia as a comparison. The authors do use GnomAD as a comparison for allele frequencies (which are high).
We intentionally selected common polymorphic variants for analysis to facilitate Hardy–Weinberg equilibrium assessment. Another important point is that we studied groups of adolescents, i.e. cohorts that have not been selected or stratified. Our study involved adolescents who studied in the same schools in the same typical district of Novosibirsk (Russia). In white populations, the frequencies of some genetic variants can vary significantly; in our previous studies, we found a difference of Russian white population and GnomaD data on the frequencies of many genes. Therefore, simply comparing our cohorts with a cohort of adolescents from another European country will not obtain new information. As a result of adaptation, cross-cultural marriages, and genetic drift, the frequencies of some variants differ not only between countries, but also within Russia (between the western and eastern regions). Therefore, in our opinion, such a comparison makes sense in the same time interval and in samples of children or adolescents in order to avoid stratification. We do not have other samples of adolescents collected in the same way, but we hope that our approach may be of interest to other researchers who will check our results.
I would recommend adding some references to define the “long-term socio-economic crisis in Russia” (Line 51).
We have added references [10] and [11].
I find the approach of selecting the candidate genes a bit bias and understand that there is probably lack of funding to support processing their cohort using genome sequencing or a very small pilot GWAS. Most of the genes that they are targeting are involved in many different genetic disorders.
Besides that, we analyzed VNTRs, which cannot be simply genotype by NGS.
In this work, we did not set out to confirm or refute the data on the association of selected polymorphic sites with certain phenotypes or diseases; it is obvious that stress-induced disorders are multifactorial, and their development is the result of G × E interaction.
Medical evaluation of the human subjects was conducted every 5 years that included monitoring of mental, physical, and emotional state. There was more information regarding the ascertainment in their previous publication
We have added information and reference, where this cohorts were discribed at the first time [74]
How does one collect honest emotional data over 5 year period, is there a regular questionnaire? Data from this is not shown in the paper regarding the mental, physical and emotional state. The phenotypic data needs to be declared; the range, average to show if the three groups differ in any way.
There is many studies on the analysis of associations of genotypes and phenotypes in people of different ages, both in the population and in patients, but this was beyond our interest. In this work, we assessed genetic trace in a population after a long crisis; we were not interested in the phenotypes of the studied sample. Information about mental, physical and emotional state was in the Methods only to characterize the study, which has been conducted at the Institute of Internal and Preventive Medicine since 1989. We have removed this phrase to avoid confusing readers.
However, what about the status of the genetic markers in animal models for anxiety and fertility?
We deliberately tried not to refer to studies conducted on model animals, although many papers have been published on the relationship of social defeat stress and subdominant reproduction behavior. The main features of the period under study in Russia were the collapse of value models, social institutions, and contradictions within public consciousness. Such factors are hardly reproducible in laboratory animals.
Without any phenotypic data, I”m not convinced that this paper should be published in Genes. Based on the data shown, it is very subjective and open to bias
We tried to conduct our research as non-subjectively as possible. The groups studied differed only in the time of birth relative to the crisis. No segregation was based on phenotypic traits.
The groups had the same (i) age, (ii) place of residence, and (iii) race. Standard genotyping was carried out using controls for contamination and controls for the reliability of the results using selective Sanger sequencing. We used 2 control groups to reduce the accidental frequency deviation.
We have expanded the Introduction section and include information about studies based on comparisons of genotypes without using phenotypes information (ref. 7 and 8). Previously, our Review (Mikhailova S.V. Problems with studying directional natural selection in humans. Vavilovskii Zhurnal Genetiki i Selektsii = Vavilov Journal of Genetics and Breeding. 2023, 27(6):684-693. DOI 10.18699/VJGB-23-79 [in Russian]) had been published, where we discuss different methods of studying natural selection in human. Unfortunately, the English version of this article is not yet available.
Reviewer 2 Report
Comments and Suggestions for Authors
This is a short report examining allele frequencies of a number of VNTRs in genes previously shown to be associated in some populations with stress response in some manner. The groups examined had been examined for other factors in the past. Overall the rationale, methods and results are sound. Only one significant difference was found in a group that remained in Hardy-Weinberg equilibrium. This outcome leads to a conclusion that populations are affected by living through a stressful time. I think this is a rather strong statement to make. The change noted indicates any response is quite limited.
The introduction presents some rationale for the VNTRs selected for further analysis. Whilst these have been shown to be significant in previous work it would be useful to know which populations have been investigated in order to determine their relevance to the current study. I am not entirely convinced that the APOE genotype will be relevant in this instance. For the allele frequency to be important, as noted in Alzheimer's disease, the analysis would have to be more associated with a clinical phenotype.
In a general sense I am interested in the phenotypic effect of these polymorphisms. Would the authors like to offer some conjecture as to whether a single change is sufficient to have any obvious effect?
Are the numbers in the different groups sufficient to reach any firm conclusions?
It is not immediately clear why a change in allele frequency would arise as a result of a stressful environment. Perhaps a little more information to support this hypothesis would be useful. The comparison data are largely from situations where the comparison was between groups showing a reduced response to stress and those not. In this instance the populations are not selected in the same manner. As indicated in the limitations this is a preliminary report which is interesting but, currently, not convincing.
Comments on the Quality of English LanguageSome slight discrepancies in the English particularly around plurality. This does not impact upon the work but could be tweaked.
Author Response
Thank you very much for your attention to our article. We hope that we were able to improve it according to your comments
English has been corrected
The introduction presents some rationale for the VNTRs selected for further analysis. Whilst these have been shown to be significant in previous work it would be useful to know which populations have been investigated in order to determine their relevance to the current study. I am not entirely convinced that the APOE genotype will be relevant in this instance. For the allele frequency to be important, as noted in Alzheimer's disease, the analysis would have to be more associated with a clinical phenotype.
In this study, we were not interested in the phenotypes of adolescents; we did not perform any selection based on phenotypes, so that the groups to be random. Common polymorphic sites that we selected for frequency estimation, in our opinion, could potentially be associated with adaptation to prolonged stressful conditions. The data presented in the Introduction section serve to justify our choice of the variants.
APOE was not selected for its association with Alzheimer's disease, but because the gene is thought to be associated with brain resilience [62–65], and differences in the effects of stress in carriers of different genotypes have been shown [66–68]. Alzheimer's disease develops in the vast majority of cases in the post-reproductive period, and therefore cannot directly affect the number of children of an individual. At the same time, lipoprotein levels are associated with learning and memory. Studies assessing natural selection in humans have also noted selection for triglyceride and cholesterol levels:
Byars S.G., et al. Colloquium papers: natural selection in a contemporary human population. Proc. Natl. Acad. Sci. USA. 2010;107(Suppl.1):1787-1792. DOI 10.1073/pnas.0906199106
Mathieson I., et al. Genome-wide patterns of selection in 230 ancient Eurasians. Nature. 2015;528(7583):499-503. DOI 10.1038/nature16152
Kuijpers Y. et al. Evolutionary trajectories of complex traits in European populations of modern humans. Front. Genet. 2022;13:833190. DOI 10.3389/fgene.2022.833190).
We have included additional explanations of the study design in the Introduction and Methods sections.
In a general sense I am interested in the phenotypic effect of these polymorphisms. Would the authors like to offer some conjecture as to whether a single change is sufficient to have any obvious effect?
The results obtained are not enough to assess the role of polymorphisms, because showing a difference in frequency, rs6557168 of the ESR1 is located in an intron, and its functional analysis has not yet been described in literature. In addition, epigenetics plays an important role in the regulation of stress response genes. For now, we can only conclude that the allele frequencies of the generation born during the long-term social stress may differ from those in the rest of the population.
Are the numbers in the different groups sufficient to reach any firm conclusions?
We analyzed only common polymorphic variants because we were limited by the cohort size that was available. We checked the obtained results using Fisher's exact 2-tailed test. As we analyzed each group, we ensured that the frequency of each allele obtained for it did not change when adding results from the next multiwell plate.
It is not immediately clear why a change in allele frequency would arise as a result of a stressful environment. Perhaps a little more information to support this hypothesis would be useful
We have added information and reference [10], [11], [13], and [14].
The comparison data are largely from situations where the comparison was between groups showing a reduced response to stress and those not. In this instance the populations are not selected in the same manner.
The data presented in the Introduction section illustrate the reasons why we selected specific variants for analysis. Of course, even if associated with certain phenotypes, such VNTRs and SNPs may not significantly affect the number of children of their carriers or may affect men and women in the opposite way, which will lead to their frequencies remaining unchanged among their children generation. This is exactly what is evident in the results we obtained. Though, we found one SNV with different frequency in groups.
As indicated in the limitations this is a preliminary report which is interesting but, currently, not convincing.
Earlier, directional natural selection was found in human populations for innate immunity genes during epidemics and for metabolic genes due to worsening living conditions (ref. [5], [7], and [8]). However, after demographic transition, the targets of selection are largely genes responsible for social adaptation and behavioral phenotypes (for example, ref [1], [2], [82], and [83] in this paper). Comparison of generations born in different periods of long-term socioeconomic crisis makes it possible to detect genetic variants that are under the pressure of natural selection in the modern urban population at times of increased social stress. We have shown that there may be genes whose variant frequencies differ between generations within the same human population. The approach we used is unusual and differs from standard medical-genetic (to identify genotype-phenotype correlations) and phylogenetic (to identify differences between populations) studies, but we believe that it is justified.
Reviewer 3 Report
Comments and Suggestions for Authors
The main question addressed by the research seems to be about understanding the impact of a prolonged socio-economic crisis on the genetic characteristics of the generation born during that period in Russia. The researchers aim to assess whether there are any significant differences in the frequencies of common gene variants associated with stress-induced disorders between adolescents born before, after, and during the crisis. This could potentially reveal which genes might be under selection pressure during times of increased social stress in urban populations.
Here are some suggestions to improve the manuscript:
- The authors mention that Group 2 corresponds to a period of socio-economic crisis in Russia. However, they do not explicitly discuss how this crisis might have impacted the genetic or health profiles of the individuals in this group. They should control for potential confounding factors related to this socio-economic context.
- The blood samples were stored at −20 °C until genomic DNA was isolated. The authors should ensure that they discuss any potential degradation of the DNA samples and its impact on the results.
- The authors should provide a justification for the chosen PCR conditions, especially the denaturation and annealing temperatures, to ensure specificity and efficiency of the amplification.
- There should be a discussion on quality control measures taken during genotyping to ensure the validity of the results.
By addressing these aspects, the authors can strengthen their methodology, control for potential confounders, and enhance the validity and reliability of their results.
Author Response
Dear Reviewer, thank you for your interest in our article and comments. We hope that we were able to improve it.
- The authors mention that Group 2 corresponds to a period of socio-economic crisis in Russia. However, they do not explicitly discuss how this crisis might have impacted the genetic or health profiles of the individuals in this group. They should control for potential confounding factors related to this socio-economic context.
We have inserted more information about the crisis and references [10], [11], [13], and [14].
In this study, we were not interested in the phenotypes of adolescents; we did not carry out any selection for phenotypes when forming cohorts, so that the groups were as random as possible. G×E interaction, including epigenetic is a topic for a separate work; we were interested in the possible genetic trace of a destructive event in a population.
We have insert more information in Introduction section
- The blood samples were stored at −20 °C until genomic DNA was isolated. The authors should ensure that they discuss any potential degradation of the DNA samples and its impact on the results.
DNA was isolated after each series of blood sample collection. The phenol-chloroform extraction method we use allows us to obtain good quality DNA, which can be successfully stored at -70 Celsius for several decades
- The authors should provide a justification for the chosen PCR conditions, especially the denaturation and annealing temperatures, to ensure specificity and efficiency of the amplification.
PCR conditions were selected for each polymorphic site using the “gradient” function on exactly the amplifier on which PCR was subsequently carried out. The average annealing temperature of the primers was provided to us by their manufacturer. The denaturation temperature is standard for our type of cyclers and master mix used. We added information to the Methods section
- There should be a discussion on quality control measures taken during genotyping to ensure the validity of the results.
DNA samples were placed in multiwell plates, and the plates were subsequently used for RT-PCR, PCR-RFLP, and ordinary PCR (for VNTR analysis). Each multiwell plate contained 2 negative controls (water instead of DNA). Several randomly selected samples were reanalyzed using the Sanger sequencing (as described in Methods section).
Samples with a low amount of DNA whose analysis results did not allow unambiguous interpretation were excluded from the study. If there had been significant contamination in the samples, we would have seen an increased number of heterozygotes for all studied variants, however, the Hardy-Weinberg disequilibrium revealed for rs17689918 and rs4522666 was associated, on the contrary, with a reduced number of heterozygotes compared to what was expected.
We used 2 control groups to reduce random frequency variations as much as possible. We have made additions and clarifications to the Methods section.
Round 2
Reviewer 1 Report
Comments and Suggestions for Authors
The authors have addressed all my concerns with a clear concise sound argument. The article has also significantly improved with inclusion of more relevant citations and background information with respect to DNA studies.
Lines 263 and 269- use "real-time PCR-HRM" instead of "real-time PCR or RT-PCR"
Author Response
Dear Reviewer! Thank you for your interest in our paper
Lines 263 and 269- use "real-time PCR-HRM" instead of "real-time PCR or RT-PCR"
Perhaps this is some kind of formatting error, there is no mention of PCR in lines 263 and 269, real-time PCR using the TaqMan technology is described in lines 229 and 347, as well as in the Supplimentary, but in all these cases this is not HRM
Reviewer 3 Report
Comments and Suggestions for Authors
I would like to thank the Authors for all improvements they have done. I have no further comments.
Author Response
Dear Reviewer!
Thank you very much for your interest in our article.